# Predictors and prevalence of labor pain management practice at West Guji zone, Southern Ethiopia, 2024

**Misgana Desalegn Menesho●\*, Girma Tufa Melesse**

Department of Midwifery, Bule Hora University Institute of Health, Bule Hora, Ethiopia

\* misgedesalegn@gmail.com

## Abstract

Childbirth is a painful experience for most women, and many require assistance with pain relief. Strategies for managing labor pain include both non-pharmacological and pharmacological intervention. Childbirth is a painful experience for almost all women. The pain that women experience during labor is affected by multiple physiological and psychosocial factors and its intensity can vary greatly. Most women in labor require pain relief. This study aimed to evaluate the practices of managing labor pain and the factors associated with them among obstetric caregivers in public health facilities in West Guji, Southern Ethiopia, in 2024. An institution-based cross-sectional study was conducted. A simple random sampling method was used to select 280 participants for the study. Data collection was conducted using a self-administered questionnaire. Descriptive statistics, along with binary and multiple logistic regressions were utilized for analysis. Adjusted odds ratios (AOR) with a 95% confidence interval were calculated to evaluate the strength of the associations, and statistical significance was determined at a p-value of less than 0.05. The Prevalence of labor pain management practices among obstetric care providers in the study location was 43.3%. The factors which were statistically significant in relation to labor pain management practices, included: holding a diploma (AOR, 0.3; 95% CI: 0.2, 0.6), knowledge of obstetric analgesia (AOR, 6.4; 95% CI: 5.8, 9.9), and the belief that pharmacologic obstetric analgesia affects labor (AOR, 2.9; 95% CI: 1.4, 6.2). Although the level was higher than in some studies, a significant gap remains, and training and knowledge are vital for improvement. The knowledge gained from holding a diploma, along with awareness of obstetric analgesia and the impact of pharmacological options on labor, were factors influencing labor pain management practices. Therefore West Guji Zone Health Department: Should create awareness concerning labor pain management practice.

**Data availability statement:** The data sets used and analyzed during the current study were uploaded as Supporting Information.

**Funding:** The authors have declared that no competing interests exist.

**Competing interests:** The author(s) received no specific funding for this work.

## Background

The changes in the female reproductive tract's structure and physiology that get the fetus and placenta ready for delivery are referred to as labor. This often occurs between 37 and 40 weeks of gestation, when the baby is fully formed at full term. The end of the baby's time in the womb and the start of the process of adjusting to life outside the mother are marked by labor [1]. Each woman may experience pain differently. Effective management of labor pain benefits both the mother and the newborn [2].

For practically all women, giving birth is a terrible process. Women's labor pain can vary widely in intensity and is influenced by a number of physiological and psychological variables [3]. The majority of laboring women need pain medication. Labor pain management techniques include non-pharmacological therapies and pharmacological interventions [4]. Even though there are resources available, many healthcare providers in low-income settings do not routinely provide effective pharmacological pain management during labor and childbirth. A study conducted in Tanzania and Ibadan, Nigeria, revealed that 39.1% of labor pain management methods were pharmacological, while 34.5% were non-pharmacological [5–7].

Giving the finest anesthetic agent available in a timely manner is only one aspect of managing pain during childbirth. Every woman should have a thorough conversation before choosing a pain management strategy. Every woman should be thoroughly informed about all of the pain management choices available in the medical facility where she will give birth before she enters labor. All healthcare professionals has an obligation to educate women about all available options, including promoting access to pain management within the medical institution [8].

Numerous factors have affected the experience of childbirth, including medical interventions, the degree of labor pain, the quality of care and services, and awareness of the labor process. Labor is sometimes viewed as a medical issue rather than a physiological phenomena as a result of medical interventions brought about by advances in medical research that aim to assure the safety of childbirth [9]. Additionally, the another study at Durame Hospital in central Ethiopia shows that a lack of procedures hindered the use of labor analgesia [10]. According to a Nigerian study, labor pain management practices were significantly correlated with educational attainment [6].

According to the results of a study done in a public hospital in Ethiopia, over 52% of medical professionals were concerned about the safety of using pharmaceutical methods to manage labor pain, 79% of participants anticipated labor pain, 38% believed it to be a normal process, 19% believed it would take longer to manage, and 17% believed it might negatively impact the unborn child [11].

Around 140 million births take place annually throughout the world, and the majority of them are vaginal deliveries among expectant mothers who have no known risk factors for complication at the start of labor [12]. Complications that occur during or shortly after childbirth, typically as a result of hemorrhage, obstructed labor, or sepsis, account for about one-third of maternal fatalities from pregnancy-related causes [13].

Both the mother and the fetus may suffer from pain. It is important to consider the psychological impacts of extreme pain, particularly when they are linked to a poor

outcome for the mother or fetus [14]. For women and children, the effectiveness and adverse effects of various pain management techniques varies. Midwives can use inhaled nitrous oxide gas without a prescription in many European nations where doctors typically prescribe it for pain management. In the UK, this gas is 53% nitrous oxide and 47% oxygen [15].

High levels of training are needed in developing nations; staff members will have little to no training in modern drug procedures, such as the use of intravenous or neuraxial opioid medications, and technology that makes labor analgesia successful. They won't know about potential difficulties and won't feel confident enough to begin these treatments. A common concern is that using opioids to treat acute pain can quickly result in addiction [16]. A Nigerian study found that the majority of participants (79.7%) were registered nurse-midwives. Among them, 90.1% used reassurance for pain relief. Another study conducted in East Gojjam, Ethiopia, revealed that being a midwife was significantly associated with labor pain management practices [17,6].

Childbirth pain is complicated and is seen as a personal experience. A flexible strategy with multiple options is necessary for pain management so that a woman can select the one that best suits her [18].

The World Health Organization (WHO) considers pain management to be a quality standard of care, emphasizes that all medical care must be given promptly, appropriately, and with respect for a woman's choice, culture, and needs [19].

Numerous factors influence how much pain a woman experiences and how well she manages it. For example, when women are well-prepared and supported, they develop endogenous analgesic chemicals that help them cope with the pain of delivery [20].

According to medical experts, hospital-related factors were the main obstacles to implementing pain-relieving techniques [21].The majority of medical practitioners are aware of both pharmacological and non-pharmacological methods of managing pain. Staffing constraints, equipment shortages, lack of availability to nitrous oxide or epidural medications, and concerns about the impact of opioids on mothers and/or children were among the difficulties [22].

According to comprehensive and focused guidelines on intrapartum care for expectant mothers and their unborn children, which have recently been combined with the recommendations of the World Health Organization, when expressed as a component of care, would ensure talent quality and fact-based care in all territorial contexts. The principle highlights unnecessary, non-fact-based, and potentially harmful intrapartum care practices that weaken mothers' innate childbearing abilities, destroy materials, and impair equity, in addition to instituting crucial clinical and non-clinical practices that aid in an enjoyable childbearing experience. In order to ensure that every proposal is accurately understood and implemented in practice, each setting-specific recommendation is precisely articulated within it, with the contributing expert providing extra information when needed [23,24].

A Tanzanian study identified several main obstacles to providing pain relief options: health system shortcomings (staffing, equipment, and protocols), a lack of education, limited opportunities for practicing pain management, and negative beliefs and practices [25].The degree of labor pain varies widely from woman to woman and has several physiological and psychological components. Pethidine is the most often used medication to alleviate pain in the labor ward [26] and Massages, walks, showers, music, and different posture approaches are examples of nonpharmacological treatment options that might assist the mother cope with the discomfort. With just education and some non-invasive pain management techniques, many women can manage their pain [27].

Pethidine may be linked to decreased breastfeeding success because of the child's diminished capacity to initiate and sustain breastfeeding, while morphine is preferred over pethidine due to its shorter half-life in women and children. The opioid is more effective in early active labor and less effective after seven centimeters of dilation [28].

According to the survey, 37.9% of women in Ethiopia utilized analgesics during childbirth. Pregnant women, family friends, and medical professionals are often concerned about experiencing the agony of childbirth again [29]. 34.4% of medical personnel in the Amhara region used pain medication during labor, according to another survey [17]. According to a study done at public health facilities in the East Gojjam Zone of the Amhara regional state, 34.4% of workers reported using a labor pain management technique. which were 4% pharmacological and 30.4%

non-pharmacological [17,30,31]. According to studies, there are a number of barriers to the use of labor analgesia, including those pertaining to patients, the system, and medical personnel. In addition to being attentive, effective pain management during labor provides significant physiological and psychological advantages for both the mother and the unborn child [32].

Pharmacological and non-pharmacological therapies are the gold standard for reducing labor and delivery pain, but their application and usefulness are limited, particularly in underdeveloped nations like Ethiopia. Therefore, the worst pain is labor pain; if labor pain is managed during a typical labor, the feto-maternal outcome and the laboring women's satisfaction are improved. This study will support policy makers, health planners and managers in drafting guidelines and provide guidance on managing pain at labor. It also helps obstetric care professionals to provide prenatal education on the need for methods to relieve pain during labor, both pharmacological and non-pharmacological methods during childbirth. It will identify the challenges of labor pain management, method to relieve pain at labor and factors influencing pain management at labor to improve institutional delivery to achieve the goal of sustainable development, reduce maternal mortality and morbidity and prevent the death of newborns through the use of services provided by qualified obstetric care givers. In addition, this study will serve as input for the future researcher. Thus, the purpose of this study was to evaluate the extent of labor pain management practices and related characteristics among obstetric care providers in West Guji zone public health facilities.

## Methods

### Ethics statement

With the ethical code BHU/IOH/020/16, the Institutional Research Ethical Review board of Bule Hora University College of Health and Medical Science granted ethical approval. The health science college's research organizing and reviewing committee gave the project the go-ahead morally. Every technique was used in accordance with the pertinent policies and procedures of Bule Hora University. Participants in the study gave written informed consent after being told of the study's goal and the confidentiality of all information; no personal information was left on the questionnaire. The consent of the participants whose age were under 18 had taken from their partner.

### Study area and study period

The study was carried out in Southern Ethiopia's Oromia Region's West Guji Zone. The West Guji zone is situated at 5°35′ N Latitude and 38°15′ E Longitude, 467 kilometers south of Addis Ababa. Nine districts and 196 kebeles make up the zone; 166 of these are rural, and 30 are urban. There are an estimated 1,389,821 people living there, with 681,012 men and 708,809 women. There are 166 health posts, 42 health centers, 2 primary hospitals, 1 general hospital, 478 health extension workers, and 860 health care providers in the West Guji Zone. The research was carried out between July 11th, 2024, and September 30th, 2024.

### Study design

An institutionally based cross-sectional study design was used.

### Source population

All medical personnel who worked in public health facilities in the West Guji Zone made up the source population.

### Study population

All obstetric care providers on staff at West Guji Zone public health facilities during the study period made up the study population.

### Inclusion criteria

Every obstetric care provider in the delivery room, including physicians, midwives, nurses, health officers, and anesthetists.

### Exclusion criteria

This study eliminated all obstetric care providers who had been hired within the previous six months.

### Sample size determination

Each objective's sample size was determined, and the ideal sample size was chosen. Using a single population proportion calculation and the following assumptions, the sample size was established for objective one: the percentage of labor pain management practices derived from a prior study conducted in Tigray General Hospitals in Northern Ethiopia.

Formula, n= $(Z\alpha/2)^2$ p (1-p)/d 2, where n= $(1.96)^2$ (0.433) (0.566)/ $(0.05)^2$ =377

Where; n = sample size

d=margin of error (5%)

p= proportion of labor pain management practice (43.3% or p=0.433 [33]).

$Z\alpha/2$= critical value at 95% CI is 1.96.

By adding 10% of non-response rate 377 + 10%=414.7 approximatly ~ 415

Sample size was also calculated for objective two using double population proportion formula by Epi-Info version 7, by using type I error 5%,power of 80% and ratio of exposed to un exposed 1:1 and 95% CI from previous study conducted in Tigray General Hospitals [33].

The sample size for both objectives was calculated using the technique above, and the largest sample size—obtained from objective 1 (415), was chosen. Since there are fewer than 10,000 people living in the research region overall, a correction formula was applied.

Hence, **n/1+ (n/N) =415/1+ (415/860) =280 subjects**

### Sampling procedure

All three public hospitals and forty-two health centers in the Zone were chosen, while fifteen health centers were chosen using a basic random selection method. Bule Hora General Hospital, Karcha Primary Hospital, and Malka Soda Primary Hospital were the hospitals chosen for the study; they had 154, 50, and 33 obstetric care providers, respectively.

A total of 294 obstetric care providers from the following health centers were chosen for this study: Q/Rasa, Garba, Ela Farda, G/Soke, Ela Dima, Dimtu, Bore, Guwanguwa, Bukisa, Finchawa, G/Dibisa, Bule Hora, Shara, Suro, and Soda. Each health facility received a proportionate share of the necessary number of study participants based on the number of obstetric professionals housed there. By proportionate allocation to sample size, 185 obstetric care providers were chosen from all hospitals, and 95 obstetric care providers were chosen from all health centers. Finally, 280 study participants were chosen using a simple random selecting procedure.

The sampling process was explained in Fig 1.

### Data collection tools

A pretested, structured, self-administered questionnaire with data collection assistance was used to gather the data. The instrument was created in English so that all respondents could understand it and was adapted from many studies while taking into account the local circumstances of the study area and the study's goal.

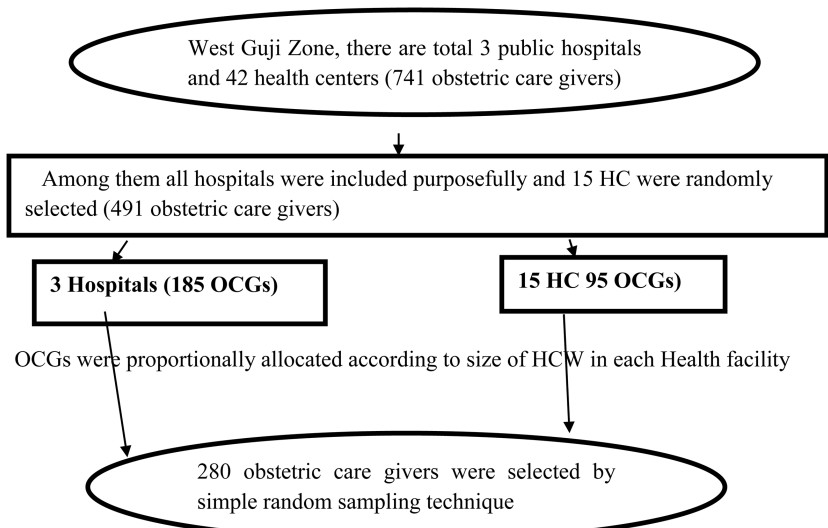

**Fig 1. Schematic representation of sampling procedure to assess labor pain management practice among health care workers of west Guji Zone, 2024.**

## Operational and term definition

**Labor:** a continuous process in which progressive regular uterine contractions result in the expulsion of the fetus.

**Obstetric care givers:** skilled health professionals who are giving maternal care service in the delivery room.

**Adequate Knowledge:** caregivers who knew greater than or equal to the mean score value of knowledge related obstetric analgesia questions.

**Inadequate Knowledge:** caregivers who knew less than the mean score value of knowledge related obstetric analgesia questions.

**Positive attitude:** respondents who answered greater than or equal to the mean score value of attitude related obstetric analgesia questions.

**Negative attitude:** respondents who answered less than the mean score value of attitude related obstetric analgesia questions.

**Good practice:** obstetric care givers who are practicing greater than or equal to the mean score value of listed labor pain management methods.

**Poor practice:** obstetric care givers who are practicing less than the mean score value of listed labor pain management methods.

## Data collection procedure

Nine assistant lecturers from Bule Hora University College of Health and Medical Sciences were hired as supervisors, and eighteen BSc midwives —one from each health facility out of data collection area —were hired as data collectors. They received two days of training on data collection. Every obstetric care provider in the delivery room received a self-administered structured questionnaire during the data collecting period, and they were asked to complete it truthfully.

### Data quality control

A systematic questionnaire was utilized to control the quality of the data, and 5% of the study population at Adola General Hospital took a pre-test to make sure the tool was clear and applicable before the main survey was conducted and modified as needed.

For two days, supervisors and data collectors received training on the study's objectives, study instruments, data gathering techniques, interviewing techniques, verifying the correctness of the questionnaires during data collection, and handling the data that was gathered.

In order to preserve the quality of the data and provide the data collectors with the appropriate feedback, the supervisor carried out his or her daily supervision, spot-checking and assessing the completed questionnaire. The primary investigator oversaw the coordination of the entire endeavor.

### Data processing and analysis

After the data was exported to SPSS version 25 for statistical analysis, it was cleaned (checked for missing values, recoding, and outliers) using Epi data 3.1. Cross tabulation was performed for additional analysis in order to obtain descriptive summary statistics and find correlations between dependent and independent variables for data exploration.

Numerical variables were represented by mean and standard deviation in descriptive statistics, whereas categorical variables were represented by frequency and percentage and shown in tables and pie charts. Histograms were used to verify that continuous data were normal.

In order to control for potential confounders and predict the relationship between explanatory variables and outcome, bivariable logistic regression analysis was utilized to evaluate the relationship between the dependent and all independent variables and to identify candidate variables for multivariable analysis at p-value <0.25.

To determine whether factors were statistically significant at $P < 0.05$, multivariable logistic regression analysis was used, taking into account the variables chosen from the bivariate logistic regression analysis. The variables' fitness was assessed using the Hosmer and Lemeshow model goodness of fit test, which was performed when the p-value was greater than 0.05. Independent variables with comparable effects on the result variable were examined for multicollinearity.

## Results

### Socio-demographic characteristics of respondents

268 of the 280 eligible obstetrics care workers participated in the survey, representing a 95.7% response rate. The mean age of the obstetric care providers was 29.14 years with SD±4.722, ranging from 20 to 50 years. Approximately half of the respondents, 157, or 58.6%, were between the ages of 20 and 29. Of the responders, 172 (64.2%) were married, making up over two thirds. One hundred and sixty-one (60.1%) of the participants were nurses, 152 (56.7%) had a degree, 199 (74.3%) were men, and 124 (46.3%) were protestants. A total of 214 individuals, or 79.9%, were Oromo.

Table 1 has provided an explanation of the remaining sociodemographic characteristics of the respondents.

### Prevalence of labor pain management practice

Data regarding the management of labor pain was gathered from the obstetric care providers. As a result, 116 (43.3%) of the total respondents provided labor pain management to patients for laboring mothers, whereas 152 (56.7%) did not.

### Knowledge of obstetric care workers related to labor pain management

Approximately 240 (89.6%) of the participants knew how to manage childbirth discomfort. Eighty-eight(40.8%) of them said they use non-pharmacological ways, 76 (31.7%) said they use both pharmacological and non-pharmacological methods, and

**Table 1. Socio-demographic characteristics of Obstetric care workers, West Guji, 2024.**

| Characteristics | Category | Number | Percentage |
|---|---|---|---|
| Residence | Urban | 189 | 70.5 |
| | Rural | 79 | 29.5 |
| | Total | 268 | 100 |
| Profession of the respondents | Medical doctor | 12 | 4.5 |
| | Anesthesia | 4 | 1.5 |
| | Nurse | 161 | 60.1 |
| | Midwife | 47 | 17.5 |
| | Health officer | 36 | 13.4 |
| | Emergency surgery | 3 | 1.1 |
| | Others | 5 | 1.9 |
| | Total | 268 | 100 |
| Qualification of respondents | Gynecologist | 4 | 1.5 |
| | MSc | 11 | 4.1 |
| | GP | 11 | 4.1 |
| | BSc | 152 | 56.7 |
| | Diploma | 90 | 33.6 |
| | Total | 268 | 100 |
| Experience of the respondents | <1year | 32 | 11.9 |
| | 1-5years | 129 | 48.1 |
| | 5-10years | 68 | 25.4 |
| | 10-15years | 30 | 11.2 |
| | >15years | 9 | 3.4 |
| | Total | 268 | 100 |
| Current position of respondent | House officer | 9 | 3.4 |
| | Medical director | 6 | 2.2 |
| | Staff member | 250 | 93.3 |
| | Hospital manager | 3 | 1.1 |
| | Total | 268 | 100 |

approximately 115 (42.9%) said they utilize back massages. Of the study participants, about 66 (27.5%) employ the pharmaceutical technique. 158 (59%) of those who used the pharmacological technique did so intramuscularly.

**Attitude of obstetric care workers related to labor pain management**

Regarding attitude, 220 (82.1%) of participants believed that labor pain was normal and that mothers must endure it. More than half (53.7%) agreed that using pain management affected the course of labor, while 145 (54.1%) suggested that all mothers should receive labor pain relief. Approximately 209 (78%) of the participants believed that pharmacologic obstetric analgesia had an impact on the results of labor and delivery; of these, 116 (55.5%) believed that it had an impact on the results of labor. Approximately 118 (43.3%) of the respondents performed labor pain management; of those who did, almost half (52.6%) applied labor pain management for episiotomy. Additionally, about half of the study participants used pharmaceutical methods.

**Institutional related factors of respondents**

The primary institutional factors that prevented respondents from regularly practicing labor pain management were: 54 (20.1%) respondents had no explanation; 145 (54.1%) study participants complained that there was no medication; 12

(4.5%) said that there was no equipment; 32 (11.9%) said that the hospital setting was inappropriate; and 13 (4.9%) said that they lacked knowledge or expertise. In a similar vein, every participant indicated that the most frequent barriers to applying the techniques to alleviate labor pain were a lack of qualified healthcare professionals, policies, and procedures.

**Factors associated with practice of labor pain management among respondent**

The bivariable analysis revealed a significant association ($p < 0.25$) between the respondents' age, sex, current residence, experience, qualification, knowledge of obstetric analgesia, types of nonpharmacological methods, belief that every mother should receive labor analgesia, and belief that pharmacologic obstetric has an effect on the labor and delivery outcomes. However, there was no association with the respondents' marital status.

Table 2 below lists the factors related to pain management practice.

According to multivariable analysis, the following factors remained statistically significant of labor pain management practice at p-value $< 0.05$: respondent qualification, obstetric analgesia knowledge, and the belief that pharmaceutical obstetric analgesia has an impact on labor and delivery outcomes (Table 2).

Obstetric care providers with a diploma were 0.3 times less likely than their peers to use labor pain treatment (AOR, 0.3; 95% CI: 0.2, 0.6).

Compared to the referent, respondents who were knowledgeable about obstetric analgesia were 9.4 times more likely to use labor pain management techniques (AOR, 6.4; 95% CI: 5.8, 9.9).

Compared to referents, obstetric care personnel were 2.9 times more likely to perform labor pain management if they had considered the impact of pharmaceutical obstetric analgesia on labor and delivery outcomes (AOR, 2.9; 95% CI: 1.4,6.2) (Table 3).

## Discussion

The scope of labor pain treatment practice and its contributing components were evaluated in this study. The respondents' qualifications, their understanding of obstetric analgesia, and their belief that pharmacologic obstetric analgesia had an impact on the results of labor and delivery were factors linked to labor pain management practices.

According to this survey, 43.3% of obstetric care personnel used labor pain control techniques. When compared to the findings of the Kembata Tembaro Zone, Southern, and East Gojjam zones, Amhara region, Ethiopia, which were 37.9% and 34.4%, respectively, this study yielded greater results [30]. The effort put forth by the medical staff in the research location and the interval between the previous study and the present one could be the cause. Beside, The methodology, the infrastructure's accessibility and availability, and the differences in the backgrounds of the study participants were all cited as potential explanations for the disparities. This disparity may also be caused by factors related to the health system, as well as the significant efforts made by health bureau and other healthcare facilities to increase public awareness. This proportion was greater than the 40.1% found in a study done in Amhara Regional State Referral Hospitals. The study's breadth may have contributed to the potential cause [34]. This study's results were lower than those of a study done in Zaria, Nigeria, which came out at 48.4%. [35]. This disparity may result from socioeconomic status differences and a lack of information about labor pain treatment practices among healthcare professionals across areas.

In this study, labor pain management practices were substantially correlated with knowledge of obstetric analgesia. Respondents in this study were 6.4 times more likely to handle labor pain if they knew about obstetric analgesia. This result is in line with research done in Ethiopia's Amhara region's East Gojam zone [17].

In the current study, labor pain management practices were substantially correlated with obstetric care personnel' diplomas. According to this study, obstetricians with diploma-level training were 70% more likely to have subpar labor pain management practices. The results of this study are consistent with research done at general hospitals in the Tigray Region of North Ethiopia and Amhara Regional State Referral Hospitals in the Northwest, respectively [34,33]. The current investigation found a substantial correlation between labor pain management practices and the knowledge

**Table 2. Bivariable and multivariable logistic regression analysis of labor pain management practice and associated factors among obstetric care workers, West Guji Zone, 2024.**

| Characteristics | Labor pain management practice | | COR(95%CI) | AOR(95%CI) |
|---|---|---|---|---|
| | **Yes** No. (%) | **No** No. (%) | | |
| **Age of respondents** | | | | |
| 20-29 | 62(53.4%) | 95(62.5%) | 0.5(0.2,1.9) | 0.8(0.2,3.6) |
| 30-39 | 48(41.4%) | 52(34.2%) | 0.8(0.2,2.7) | 0.9(0.2,4.5) |
| =>40 | 6(5.2%) | 5(3.3%) | 1 | |
| **Sex of respondents** | | | | |
| Male | 93(80.2%) | 106(69.7%) | 0.6(0.32,1.01) | 1.6(0.8,3.1) |
| Female | 23(19.8%) | 46(30.3%) | 1 | |
| **Residence** | | | | |
| Urban | 91(78.4%) | 98(64.5%) | 2.0(1.2,3.5) | 1.7(0.9,3.2) |
| Rural | 25(21.6%) | 54(35.5%) | 1 | |
| **Experience of respondents** | | | | |
| <1 year | 10(8.6%) | 22(14.5%) | 1 | 1.4(0.24,8.4) |
| 1-5Years | 55(47.4%) | 74(48.7%) | 0.9(0.2,4.4) | 2.7(0.6,13.4) |
| 5-10Years | 38(32.8%) | 30(19.7%) | 1.5(0.4,6.2) | 2.4(0.5,12.3) |
| 10-15Years | 10(6.8%) | 20(13.2%) | 2.5(0.6,10.9) | 2.3(0.4,13.9) |
| >15 years | 3(2.6%) | 6(3.9%) | 1.0(0.2,4.8) | |
| **Qualification of respondents** | | | | |
| Gynecologist | 1(0.9%) | 3(2%) | 1 | 0.9(0.3,2.7) |
| MSc | 3(2.6%) | 8(5.3%) | 0.2(0.02,2.12) | 1.8(0.9,3.5) |
| GP | 7(6.0%) | 4(2.6%) | 0.24(0.6,0.96) | 1.6(0.7,3.6 |
| BSc | 60(43.1%) | 102(67.1%) | 1.1(0.3,4.1) | **0.3(0.2,0.6)*** |
| Diploma | 55(47.4%) | 35(23%) | 0.3(0.2,0.54) | |
| **Knowledge of obstetric analgesia** | | | | |
| Yes | 113(97.4%) | 127(83.6%) | 7.4(2.2,25.2) | **6.4(5.8,9.9)*** |
| No | 3(2.6%) | 25(16.4%) | 1 | |
| **Types of non-pharmacological method** | | | | |
| Acupuncture | 3(2.6%) | 2(1.3%) | 1 | |
| Diversional therapy | 5(4.3%) | 12(7.9%) | 4.8(0.64,15.5) | 7.9(0.8,8.9) |
| Psychotherapy | 51(41.4%) | 51(33.6%) | 1.3(0.3,5.3) | 1.7(0.4,7.8) |
| Massage the back | 49(42.2%) | 66(43.4%) | 2.9(1.1,8.1) | 3.5(1.2,10.1) |
| Hypnosis | 1(0.9%) | 1(0.7%) | 2.4(0.9,6.3) | 1.8(0.6,5.2) |
| Show the pt to bear down | 4(3.4%) | 1(0.7%) | 3.2(0.2,28.7) | 1.8(0.1,3.6) |
| Help to do labor exercises | 6(5.2%) | 19(12.5%) | 12.7(1.2,13.6) | |
| **Thought of every mother should receive labor analgesia** | | | | |
| Yes | 74(63.8%) | 71(46.7%) | 2.0(1.2,3.3) | 1.7(0.9,2.9) |
| No | 42(36.2%) | 81(53.3%) | 1 | |
| **Thought of pharmacologic obstetric analgesia has effect on the labor and delivery outcomes** | | | | |
| Yes | 101(87.1%) | 108(71.1%) | 2.7(1.4,5.2) | **2.9(1.4,6.2)*** |
| No | 15(12.9%) | 44(28.9%) | 1 | |

**Note: * shows statistically significant at P value<0.05.**

that pharmaceutical obstetric analgesia had an impact on labor and delivery outcomes. The possible justification for this finding could be respondents who were diploma might not be know much about labor pain management practice and services. instead, they were more interested in procedural management. Conversely, obstetric healthcare providers with degree and higher levels of education were better informed about health issues and knew how to manage the labor pain

**Table 3. Abbreviation and Acronyms.**

| AOR | Adjusted odds ratio |
|---|---|
| HH | Household |
| HCP | health care provider |
| MCH | Maternal and child health |
| MMR | Maternal mortality rate |
| MOH | Ministry of Health |
| PNC | Postnatal care |
| NGO | Non-Governmental Organization |
| SPSS | Statistical package for social science |
| SRS | simple random sampling |
| SS | Sample size |
| TTBA | trained traditional birth attendants |
| WHO | World Health Organization |

in multidimensional ways.in fact, with increasing the academic rank the possibility of advanced procedural accomplishment tend to increase proportionaly.

In comparison to the referents, obstetric care personnel who believed that pharmacologic obstetric analgesia had an impact on labor and delivery outcomes were 2.9 times more likely to use labor pain management techniques. The results of this study are similar to those of a study carried out in Ethiopia's Amhara region at the University of Gonder [36]. It may be challenging to determine the causal association between the research variables due to the cross-sectional study methodology used in this investigation.

The study lacks the depth to determine the elements that contribute to early prenatal care booking because it relied on a quantitative data collection method. There was some incompleteness because the data was collected via a self-administered questionnaire. According to the study's findings, the authors advise that labor pain management procedures be improved in all medical facilities; providers should receive the necessary training, and the significance of managing labor pains during labor and delivery should be emphasized in order to raise the standard of delivery care.

## Conclusion

The study found that a significant proportion of obstetric care providers did not practice labor pain management. The practice of labor pain management was significantly influenced by the providers' qualifications, their knowledge of obstetric analgesia, and their beliefs about the impact of pharmacologic analgesia on labor and delivery outcomes. Therefore West Guji Zone Health Department: Should create awareness concerning labor pain management practice and strengthening both pharmacological and nonpharmacological labor pain management practice at Obstetric ward for all health care workers and build the capacity of health care provider to practice it and provide necessary services as well as excel the health professions quality to practice labor pain during delivery.

## Supporting information

**S1 Data. The data sets used and analyzed for study.**
(SAV)

## Acknowledgments

We would also like to express our gratitude to our lecturers for providing pertinent knowledge from the beginning of the title to the finished product.

## Author contributions

**Conceptualization:** Misgana Desalegn Menesho, Girma Tufa Melesse.

**Data curation:** Misgana Desalegn Menesho, Girma Tufa Melesse.

**Formal analysis:** Misgana Desalegn Menesho, Girma Tufa Melesse.

**Funding acquisition:** Misgana Desalegn Menesho, Girma Tufa Melesse.

**Investigation:** Misgana Desalegn Menesho, Girma Tufa Melesse.

**Methodology:** Misgana Desalegn Menesho, Girma Tufa Melesse.

**Project administration:** Misgana Desalegn Menesho.

**Resources:** Misgana Desalegn Menesho.

**Software:** Misgana Desalegn Menesho.

**Supervision:** Misgana Desalegn Menesho.

**Validation:** Misgana Desalegn Menesho.

**Visualization:** Misgana Desalegn Menesho.

**Writing – original draft:** Misgana Desalegn Menesho.

**Writing – review & editing:** Misgana Desalegn Menesho.

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
