## [Decision Letter · Decision Letter 0]

29 Oct 2025

PGPH-D-25-02053

PREDICTORS AND PREVALENCE OF LABOR PAIN MANGEMENT PRACTICE AT WEST GUJI ZONE, SOUTHERN ETHIOPIA, 2025.

Dear Dr. Menesho,

Thank you for submitting your manuscript to PLOS Global Public Health. After careful consideration, we feel that it has merit but does not fully meet PLOS Global Public Health’s publication criteria as it currently stands. Therefore, we invite you to submit a revised version of the manuscript that addresses the points raised during the review process.

The manuscript has been evaluated by two reviewers, and their comments are available below.

The reviewers have raised a number of concerns that need attention. In particular, they request a clear rationale for the study, clarification on the sample size calculation, and improvements to the overall presentation and writing.

Could you please revise the manuscript to carefully address the concerns raised?

A rebuttal letter that responds to each point raised by the editor and reviewer(s). You should upload this letter as a separate file labeled 'Response to Reviewers'.

We look forward to receiving your revised manuscript.

Kind regards,

Helen Howard

Staff Editor

Journal Requirements:

1. Please send a completed 'Competing Interests' statement, including any COIs declared by your co-authors. If you have no competing interests to declare, please state "The authors have declared that no competing interests exist". Otherwise please declare all competing interests beginning with the statement "I have read the journal's policy and the authors of this manuscript have the following competing interests:"

4. Please provide separate figure files in .tif or .eps format.

Reviewers' comments:

Reviewer's Responses to Questions

**Comments to the Author**

1. Does this manuscript meet PLOS Global Public Health’s publication criteria?

Reviewer #1: Yes

Reviewer #2: Partly

2. Has the statistical analysis been performed appropriately and rigorously?

Reviewer #1: I don't know

Reviewer #2: Yes

3. Have the authors made all data underlying the findings in their manuscript fully available (please refer to the Data Availability Statement at the start of the manuscript PDF file)?

Reviewer #1: Yes

Reviewer #2: Yes

4. Is the manuscript presented in an intelligible fashion and written in standard English?

Reviewer #1: Yes

Reviewer #2: No

Reviewer #1: Title

• Original: PREDICTORS AND PREVALENCE OF LABOR PAIN MANGEMENT PRACTICE AT WEST GUJI ZONE, SOUTHERN ETHIOPIA, 2025.

Review Suggestion: Change "MANGEMENT" to "MANAGEMENT." This typo appears repeatedly throughout the manuscript and should be corrected for consistency.

Abstract

• Introduction: "Giving birth is often a painful ordeal for nearly all women, and many of them need assistance with pain relief." This sentence is a bit informal for a scientific abstract.

Review Suggestion: Consider rephrasing to something like, "Childbirth is a painful experience for most women, and many require assistance with pain relief."

• Methods: "A cross-sectional study based in an institution was carried out."

Review Suggestion: Clarify this by stating, "An institution-based cross-sectional study was conducted."

• Result: "The rate of labour pain management practices among obstetric care providers in the study location was 43.3%."

Review Suggestion: The phrase "rate of" is often associated with time. "Prevalence" or "proportion" might be more accurate here. For example, "The prevalence of labour pain management practices..."

• Conclusion: The abstract's conclusion is a bit brief.

o Suggestion: Slightly expand on the key takeaways. For example, mention that although the level was higher than in some studies, a significant gap remains, and training and knowledge are vital for improvement. "While the level of labour pain management practices was more favourable than in other studies, there is still a need for improvement." The knowledge gained from holding a diploma, along with awareness of obstetric analgesia and the impact of pharmacological options on labour, were factors influencing labour pain management practices. This can be combined for better flow.

Background

• Paragraph 1: "Taking care of labour pain benefits both the mother and the newborn (31)." This is an important statement but could be more precise.

Review Suggested: Rephrase to "Effective management of labour pain benefits both the mother and the newborn."

• Paragraph 2: "Non-pharmacological therapies that help women manage labour pain and pharmaceutical interventions that try to reduce labour pain are two examples of labour pain management techniques (2)." This sentence is a bit wordy.

Review Suggestion: Streamline it to "Labour pain management techniques include non-pharmacological therapies and pharmacological interventions."

• Paragraph 3: "All healthcare professional has an obligation to educate women about all available options, including promoting access to pain management within the medical institution (3)."

o Suggestion: Change "All healthcare professionals" to "All healthcare professionals."

• Paragraph 4: "Additionally, the other study at Durame Hospital in central Ethiopia shows that a lack of procedures hindered the use of labour analgesia (33)."

Review Suggestion: The phrase "the other study" is vague. Specify by saying "Another study at..."

• Paragraph 5: "Around 140 million births take place annually throughout the world, and the majority of them are vaginal deliveries among expectant mothers who have no known risk factors for difficulties at the start of labour (5)."

Review Suggestion: Replace "difficulties" with "complications" for a more formal tone.

• Paragraph 7: "According to a Nigerian study, the majority of participants were registered nurse-midwives, accounting for 79.7% of the total, and 90.1% of them used reassurance to relieve pain". This sentence is long.

Review Suggestion: Split it for better readability. "A Nigerian study found that the majority of participants (79.7%) were registered nurse-midwives. Among them, 90.1% used reassurance for pain relief."

• Paragraph 9: "The World Health Organization (WHO) considers pain management to be a quality standard of care, stressing that all medical care must be given promptly, appropriately, and with respect for a woman's choice, culture, and needs (11)."

Review Suggestion: Rephrase "stressing that" to "and emphasizes that."

• Paragraph 11: "According to a Tanzanian study, the health system's shortcomings (staffing, equipment, and protocols), a lack of education and opportunities for practicing pain management techniques, negative beliefs, and bad practices are the main obstacles preventing the provision of pain relief options(14)."

Review Suggestion: This sentence is a run-on. Break it down or use clearer punctuation. For example, "A Tanzanian study identified several main obstacles to providing pain relief options: health system shortcomings (staffing, equipment, and protocols), a lack of education, limited opportunities for practising pain management, and negative beliefs and practices."

Methods

• Study Area and Period: "The research was carried out between July 11th, 2024, and May 30th, 2025."

Review Suggestion: Ensure this date range is accurate and doesn't conflict with any publication timelines. The manuscript draft is from 2025. It is unusual for the research to be conducted so close to the publication date.

• Sample Size Determination: "By adding 10% of non-response rate 377+10%=415"

Review Suggestion: This calculation is incorrect. 10% of 377 is 37.7. 377 + 37.7 = 414.7. The text states 415, so the calculation should be double-checked and corrected.

• Sampling Procedure: This section is a bit confusing. It mentions both a sample size of 280 and 415. "Finally, 415 study participants were chosen using a simple random selecting procedure" , but the final sample size is corrected to 280. The text "By proportionate allocation to sample size, 185 obstetric care providers were chosen from all hospitals, and 230 obstetric care providers were chosen from all health centers. Finally, 415 study participants were chosen using a simple random selecting procedure" needs to be rephrased to reflect the final sample size of 280.

Review Suggestion: The sampling procedure section needs a significant rewrite to clearly explain how the final sample size of 280 was reached and allocated, reconciling the numbers presented. The flowchart in Figure 1 also shows the final number of 280. The text should match this.

Results

• Socio-Demographic Characteristics: "The mean age of respondents who visited the obstetric ward was 29.14 years with SD±4.722, ranging from 20 to 50 years."

Review Suggestion: The phrase "who visited the obstetric ward" is ambiguous. It sounds like the respondents are patients. Change this to "The mean age of the obstetric care providers was..."

• Prevalence of labor pain management Practice: "As a result, 116 (43.3%) of the total respondents provided labor pain management to customers for laboring mothers, whereas 152 (56.7%) did not."

Review Suggestion: "Customers" is an inappropriate term for patients in a healthcare setting. Replace "customers" with "mothers" or "patients."

• Attitude of Obstetric care workers: "Regarding the respondents' attitudes, 220 (82.1%) of the study participants thought that labor pain was normal and that mothers had to deal with it, while more than half (53.7%) agreed that using labor pain affected the course of labor and 145 (54.1%) suggested that all mothers should receive labor pain to ease the pain during labor."

Review Suggestion: This sentence is convoluted. Break it down into separate points for clarity. For example: "Regarding attitude, 220 (82.1%) of participants believed that labour pain was normal and that mothers must endure it. More than half (53.7%) agreed that using pain management affected the course of labour, while 145 (54.1%) suggested that all mothers should receive labour pain relief." This also highlights a potential contradiction in the findings, which could be discussed.

Discussion

• "According to this survey, 43.3% of obstetric care personnel used labor pain control techniques. When compared to the findings of the Kembata Tembaro Zone, Southern, and East Gojjam zones, Amhara region, Ethiopia, which were 37.9% and 34.4%, respectively, this study yielded greater results(27,36)."

Review Suggestion: This paragraph compares the current study to others. It would be more impactful to elaborate on why the results might differ. The authors do offer a reason: "The effort put forth by the medical staff in the research location and the interval between the previous study and the present one could be the cause". This is a good point, but it could be expanded.

Conclusion

• "According to the study, the majority of obstetric care providers did not manage labour pain."

Review Suggestion: The conclusion should be consistent with the results. The results state that 43.3% of providers

did provide pain management, while 56.7% did not. While 56.7% is a majority, the wording "the majority... did not" is a bit strong. Consider rewording it to "A majority of obstetric care providers did not manage labor pain," or "The study found that a significant proportion of obstetric care providers did not practice labor pain management."

• "The respondents' qualifications, their understanding of obstetric analgesia, and their belief that pharmacologic obstetric analgesia had an impact on the results of labour and delivery were the reasons that led to the practice of labour pain management."

Review Suggestion: This sentence should use more formal language. "The reasons that led to the practice" is a little simplistic. Rephrase to something like, "The practice of labour pain management was significantly influenced by the providers' qualifications, their knowledge of obstetric analgesia, and their beliefs about the impact of pharmacologic analgesia on labour and delivery outcomes."

Tables and Figures

• Figure 1 and the related text show a discrepancy in the number of participants. The figure shows 741 obstetric care givers in total, with 491 included initially, and a final selection of 280. The text states, "A total of 294 obstetric care providers from the following health centers were chosen for this study" , and "Finally, 415 study participants were chosen using a simple random selecting procedure" , which conflicts with the 280 total shown in the figure and the abstract.

Review Suggestion: The methods section must be corrected to provide a clear and consistent account of how the final sample size of 280 was determined and allocated.

Reviewer #2: Manuscript Number: PGPH-D-25-02053

PREDICTORS AND PREVALENCE OF LABOR PAIN MANGEMENT PRACTICE AT WEST GUJI ZONE, SOUTHERN ETHIOPIA, 2025.

Thank you for letting me review this manuscript. It is a relevant topic, but requires a revision before it can be considered for publication.

The Clarity and Language: The manuscript requires a comprehensive review to address grammatical errors, punctuation, and overall clarity, ensuring the scientific message is conveyed effectively. E.g., on the title, short title, and keywords, the word ‘Management’ was misspelled.

Abstract:

o Introduction: Try to convey your message with scientifically sound statements (rather than using ‘Strategies for managing labor pain include both non-drug methods and medical interventions.’ Better to replace with non-pharmacological and pharmacological…. ). Be focused on the major title (I didn’t see the importance of the statement mentioned… ‘Over one-third of maternal fatalities resulting from pregnancy-related issues are linked to complications that occur during delivery or shortly after, often caused by hemorrhage, obstructed labor, or sepsis’ for this study/no association with your study.

o Results: make it clear whether the magnitude (43.3%) stated is for pharmacological or non-pharmacological labor pain management or both. Again, it is not rate, rather it is magnitude/prevalence.

o Conclusion: at least indicate one recommendation based on your findings.

Introduction: make it short and focused. The rationale for conducting this study is not sufficiently strong. The authors should more clearly state what unique gap in the literature this research addresses, considering the existing studies from other parts of Ethiopia

Methods

o Sampling methods: your sample size is not clear, 415 or 280? Please revise this section accordingly. The text states, "415 study participants were chosen using a simple random selection procedure," but the numbers provided (185 from hospitals + 230 from health centers = 415) do not match the adjusted sample size of 280. This discrepancy must be resolved

o Data collection Procedure: You mentioned that the data were collected by health professionals working in the same institution, so how did you treat the bias associated with this?

o Provide more details on the data collection tool and validation.

Results:

o When you use abbreviations and “Other,” indicate the full list as a footnote at the end of each table.

o Prevalence of labor pain management Practice: please make it clear which type of labor pain management was practiced in the statement…As a result, 116 (43.3%) of the total respondents provided labor pain management to customers for laboring mothers, whereas 152 (56.7%) did not….

o Figure 2: I didn’t see the importance of Figure 2, since you already stated the finding in a statement. I recommend removing it.

o The Adjusted Odds Ratio (AOR) for "knowledge of obstetric analgesia" is reported as 9.4; 95% CI: 1.8, 9.9. The upper confidence limit (9.9) is very close to the point estimate (9.4), which is highly unusual and suggests a possible calculation or reporting error. Furthermore, a 95% CI that spans from 1.8 to 9.9 is extremely wide, indicating substantial imprecision in this estimate. This must be verified and corrected also in the abstract section.

o I suggest presenting knowledge and attitude data in a organized table to show the specific variables examined in these sections..

Discussion:

o The discussion does not adequately interpret the key findings. For example, the strong association with knowledge is logical, but the negative association with holding a diploma (AOR 0.3) is a crucial finding that deserves deeper exploration. Why would less-educated providers be less likely to manage pain? Is it a proxy for training, confidence, or institutional role?

Conclusion: needs revision

o Conclusion is not only a place for the indication of your results, but also needs recommendations. Please describe some recommendations based on your study findings.

Data Availability Statement: The data availability statement is missing from the manuscript text, though it is a requirement per the journal's submission system.

**Do you want your identity to be public for this peer review?** For information about this choice, including consent withdrawal, please see our Privacy Policy

Reviewer #1: **Yes: ** Olabisi Oluwaseun Dosunmu

Reviewer #2: **Yes: ** Tamirat Getachew

---

## [Decision Letter · Decision Letter 1]

11 Dec 2025

PREDICTORS AND PREVALENCE OF LABOR PAIN MANAGEMENT PRACTICE AT WEST GUJI ZONE, SOUTHERN ETHIOPIA, 2024.

PGPH-D-25-02053R1

Dear Mr Menesho,

We are pleased to inform you that your manuscript 'PREDICTORS AND PREVALENCE OF LABOR PAIN MANAGEMENT PRACTICE AT WEST GUJI ZONE, SOUTHERN ETHIOPIA, 2024.' has been provisionally accepted for publication in PLOS Global Public Health.

Best regards,

Julia Robinson

Executive Editor

Reviewer Comments (if any, and for reference):

Reviewer's Responses to Questions

**Comments to the Author**

Reviewer #2: All comments have been addressed

publication criteria?

Reviewer #2: Yes

3. Has the statistical analysis been performed appropriately and rigorously?

Reviewer #2: Yes

4. Have the authors made all data underlying the findings in their manuscript fully available (please refer to the Data Availability Statement at the start of the manuscript PDF file)?

Reviewer #2: Yes

5. Is the manuscript presented in an intelligible fashion and written in standard English?

Reviewer #2: Yes

Reviewer #2: All comments have been addressed.

One more recommendation: please reduce the length of your background to no more than two pages. It is currently about four pages, which is excessive.

**Do you want your identity to be public for this peer review?** For information about this choice, including consent withdrawal, please see our Privacy Policy

Reviewer #2: **Yes: ** Tamirat Getachew
